# Training the Vessels: Molecular and Clinical Effects of Exercise on Vascular Health—A Narrative Review

**DOI:** 10.3390/cells12212544

**Published:** 2023-10-30

**Authors:** Karsten Königstein, Konstantina Dipla, Andreas Zafeiridis

**Affiliations:** 1Department of Sport, Exercise and Health, Division Sports and Exercise Medicine, University of Basel, 4052 Basel, Switzerland; 2Laboratory of Exercise Physiology and Biochemistry, Department of Physical Education and Sports Science at Serres, Aristotle University of Thessaloniki, 62100 Serres, Greece; zafeirid@phed-sr.auth.gr

**Keywords:** exercise, hypertension, blood pressure, cardiovascular disease, microvascular function, exercise prescription

## Abstract

Accelerated biological vascular ageing is still a major driver of the increasing burden of cardiovascular disease and mortality. Exercise training delays this process, known as early vascular ageing, but often lacks effectiveness due to a lack of understanding of molecular and clinical adaptations to specific stimuli. This narrative review summarizes the current knowledge about the molecular and clinical vascular adaptations to acute and chronic exercise. It further addresses how training characteristics (frequency, intensity, volume, and type) may influence these processes. Finally, practical recommendations are given for exercise training to maintain and improve vascular health. Exercise increases shear stress on the vascular wall and stimulates the endothelial release of circulating growth factors and of exerkines from the skeletal muscle and other organs. As a result, remodeling within the vascular walls leads to a better vasodilator and -constrictor responsiveness, reduced arterial stiffness, arterio- and angiogenesis, higher antioxidative capacities, and reduced oxidative stress. Although current evidence about specific aspects of exercise training, such as F-I-T-T, is limited, and exact training recommendations cannot be given, some practical implications can be extracted. As such, repeated stimuli 5–7 days per week might be necessary to use the full potential of these favorable physiological alterations, and the cumulative volume of mechanical shear stress seems more important than peak shear stress. Because of distinct short- and long-term effects of resistance and aerobic exercise, including higher and moderate intensities, both types of exercise should be implemented in a comprehensive training regimen. As vascular adaptability towards exercise remains high at any age in both healthy individuals and patients with cardiovascular diseases, individualized exercise-based vascular health prevention should be implemented in any age group from children to centenarians.

## 1. Background

Accelerated biological vascular ageing (early vascular ageing) leads to early target organ damage and subsequently to diseases, such as atherosclerosis, chronic kidney disease, degenerative brain disease, and retinopathy [1]. The prevalence of early vascular ageing in a population, i.e., defined as aortic pulse wave velocity >90th percentile, can be as high as 37% [2], which is why the maintenance of optimal vascular health throughout the entire lifespan is already accepted as a main goal of preventive health care [3]. Regular physical activity and exercise can effectively attenuate the progression of early vascular ageing and should therefore be part of any treatment efforts aiming to improve vascular and general health [4,5]. However, the effectiveness of exercise interventions on biomarkers of vascular health is often limited and not sustainable in the long-term after the discontinuation of exercise training. Targeted exercise, based on objective and subjective individual criteria, carries the potential to overcome these limitations [6,7]. Accordingly, a better understanding of the molecular mechanisms underlying the acute responses and long-term adaptations of the vascular organ to exercise will lead to the design of more effective exercise programs, with better adherence by participants to lifelong training, and potentially will result in a better clinical outcome. Acute hemodynamic adaptations to exercise lead to changes in both endothelial and vascular smooth muscle cell function. Regular exercise training improves nitric oxide (NO) availability and reduces vascular oxidative stress [8]. As a result, endothelial vasodilator and -constrictor responsiveness as well as pulse wave velocity and arterial stiffness are improved [9]. In addition, exercise induces vascular structural adaptations and stimulates angiogenesis and arteriogenesis. This narrative review summarizes current knowledge about the acute molecular responses to exercise within the endothelial and smooth muscle cells of the vascular wall and the associated acute and chronic clinical vascular adaptations. It further addresses how training frequency, intensity, volume, and type influence these processes. Finally, practical recommendations are given for exercise training to maintain and improve vascular health.

## 2. Acute Molecular Effects of Exercise on Blood Vessels

During exercise, the increased blood flow stresses the vasculature and increases luminal shear stress. In turn, shear stress causes the deformation of glycocalyx receptors on the lumen of endothelial cells, activating calcium channels. Calcium signaling leads to prostaglandin release and consequently induces cyclic adenosine monophosphate mediated smooth muscle relaxation. Luminal shear stress activates protein kinase B, inducing endothelial NO synthase phosphorylation and consequently NO release. NO interacts with guanylate cyclase, activating a cascade of molecular signaling which leads to reduced intracellular calcium and increased intracellular potassium, favoring cell membrane hyperpolarization and smooth muscle relaxation. In addition, when NO diffuses to the luminal side of the endothelial layer, it also exerts an antithrombotic action by inhibiting platelet adhesion and aggregation [10] (Figure 1).

Acute bouts of exercise also enhance proangiogenic stimuli, such as the vascular endothelial growth factor (VEGF), which in turn signals capillary development. Previous studies have shown VEGF increases in response to a single exercise bout [11]. The level of hypoxia seems to upregulate the VEGF response [12,13]. On the other hand, acute bouts of exercise also increase the release of angiostatic factors (factors that prevent angiogenesis) such as angiostatin and platelet factor 4 (PF-4). Although there is a concomitant increase in opposing angiogenic and angiostatic factors during an acute bout of exercise, repeated exercise stimuli lead to a decrease in the angiostatic factors, increasing the abundance of angiogenic factors, promoting angiogenesis (Figure 1).

## 3. Chronic Adaptations of Blood Vessels to Exercise

### 3.1. Long-Term Structural and Functional Vascular Adaptations in Response to Regular Exercise Training

It is well established that regular exercise training improves vascular function in both health and disease. Chronic training-related adaptations of the cardiovascular system improve tissue perfusion and nutrient exchange at rest and result in a greater capacity to increase them during exercise. In this context, both structural and functional adaptations of the blood vessels have been reported (Figure 2).

Briefly, exercise stimulates angiogenesis, the formation of capillary networks, and arteriogenesis, the growth of preexistent collateral arterioles. Early reports showed a greater arteriolar and capillary density in trained individuals compared to untrained aged-matched controls [14]. An increase in capillary density in response to exercise training has also been reported in healthy individuals as well as in those with chronic disease [15]. Adaptations seem to be dependent on the fiber type and the individual’s training status. Specifically, although skeletal muscles containing mostly slow-twitch oxidative fibers have a greater capillary density [14], fast oxidative–glycolytic and fast–glycolytic muscle fibers appear to exhibit a greater increase in arteriolar density in response to training [15]. Recent evidence suggests that stimulating angiogenesis promotes healthy ageing and extends life span [16]. However, while in animal models angiogenesis can be evident after a few repeated bouts of exercise, in humans, this process might take weeks to months of regular training [17].

Studies have also suggested that repeatedly elevated shear stress, as it may occur during exercise training, induces increases in the arterial luminal diameter [18]. In detail, larger conduit arteries (increased conduit artery size in epicardial arteries and those supplying skeletal muscle) were reported in athletes compared to untrained healthy individuals [19]. Between-limb studies in athletes demonstrated higher arterial diameters and lower arterial wall thickness in the predominantly used limbs (carotid, femoral, and popliteal and brachial intima-media thickness) [18,20,21]. As a result, peak limb blood flow responses were found to be enhanced also in comparison with nonathletes, suggesting greater vasodilator reserves in athletes and trained individuals [19].

Functional vascular adaptations in response to exercise training are also evident in healthy individuals and in individuals with chronic diseases [17,22]. Alterations in the arterial tone and reductions in resting blood pressure are the results of a complex interplay of humoral, paracrine, and neural mechanisms, in which a higher bioavailability of endothelial NO as well as transiently higher levels of vascular and interstitial cell adhesion molecules, endothelin-1, and reactive oxygen species mediate the exercise stimulus. Furthermore, some studies reported that in individuals with vascular dysfunction (older adults, hypertensives), exercise training potentiate the release of NO-independent vasodilator substances such as prostacyclin I2 (in muscle and interstitial tissue) and adenosine (interstitial tissue) and to reduces vasoconstrictors such as endothelin-1 and angiotensin II [23,24]. In addition, exercise improves endothelial function through enhancing endothelial cell proliferation, inhibiting apoptosis, and mobilizing endothelial progenitor cells [25].

Collectively, all these adaptations improve constriction and dilatation responses of the vascular smooth muscle cells to vasoactive factors, increase arterial compliance, and reduce the probability of flow-limiting stenosis when cardiovascular and/or metabolic disease are present. Exercise also increases the content of collagen and elastin of atherosclerotic plaques stabilizing plaque lesions and attenuating vascular stiffening/calcification.

In addition to these macrovascular effects of exercise, exerkines (signaling molecules such as proteins, metabolites, and extracellular vesicles) are released in response to acute and/or chronic exercise from the working skeletal muscle and the cardiometabolic, nervous, and immune systems and induce adaptations in the microvascular system [26]. For example, skeletal muscular VEGF and angiopoietin 1 enhance neovascularization and angiogenesis. Further exerkine-related vascular effects are improved blood flow, endothelial function, and blood pressure.

### 3.2. The Role of Mitochondria in Exercise-Induced Vascular Benefits

An increasing amount of evidence suggests a critical role of mitochondria in the exercise-induced benefits on the vascular smooth muscle cells. However, direct evidence is sparse and most of the available evidence is extrapolated from observation in cardiac and skeletal myocytes [27]. Mitochondria are sophisticated and dynamically responsive organelles involved in energy production, redox balance, signal transduction, apoptosis (programmed cell death), and many other biological processes. Interestingly, cardiac mitochondria are different from vascular mitochondria in terms of morphology, function, and regulation. In cardiomyocytes, mitochondria comprise ~35% of the cell volume due to the high metabolic demand and oxidative capacity of the cardiac muscle. However, mitochondria in the vascular wall, mainly smooth muscle cells, comprise only ~2–5% of the tissue volume since these cells exhibit high glycolytic rates even under well-oxygenated conditions [28].

Exercise-induced adaptations involve improved mitochondrial biogenesis, signaling transduction, respiratory function, and reduced chronic oxidative stress [29]. Additionally, the de novo mitochondria combine their membranes with existing mitochondria (fusion), facilitating the distribution of metabolites, proteins, mtDNA, and other molecules to increase the capacity of ATP synthesis and improve the oxidative processes. By the process of fission, damaged areas of mitochondria are removed, and metabolic homeostasis is maintained.

Exercise promotes the degradation of damaged mitochondria via mitophagy [30,31]. Mitophagy is of great importance for the maintenance of vascular health since a growing body of evidence suggests an involvement of impaired mitophagy to the pathogenesis of vascular disease and the ageing process [29,32]. Specifically, an imbalance in the mitochondrial dynamics results in vascular dysfunction, as shown in perturbations that increase fission or decrease fusion [33,34]. Furthermore, mitochondrial fragmentation and respiratory chain dysfunction, driven by an altered activity of the GTPase dynamin-related protein-1, might contribute to vascular dysfunction [34]. A previous study showed that aerobic exercise training decreased dynamin-related protein-1 phosphorylation and increased fat oxidation and insulin sensitivity in obese insulin-resistant adults, suggesting that lifestyle-mediated improvements in substrate metabolism may be regulated through the decreased activation of the dynamin-related protein-1 [35].

In summary, chronic exercise improves mitochondrial content and dynamics, resulting in a better oxidative capacity. Although acute exercise bouts upregulate ROS generation, chronic regular exercise results in lower chronic oxidative stress and less inflammation.

## 4. Clinical Vascular Effects of Exercise

### 4.1. Effects of Exercise on Vascular Health

Regular exercise reduces the risk of cardiovascular morbidity and mortality by up to 44% [36], delays all-cause mortality [37], and increases health span [38]. Surprisingly, less than 50% of this effect is explained by reduced oxidative stress and subclinical inflammation due to modifications in cardiovascular risk factors, such as blood pressure, insulin sensitivity, blood lipids, and body composition [39]. The remaining gap in explaining vascular benefits of exercise is likely to be filled by direct effects of exercise on the vascular wall that translate into higher nitric oxide availability and favorable structural remodeling [18]. The result is a delayed progression of endothelial dysfunction and arterial stiffening with ageing. This assumption is supported by a considerable body of evidence indicating a positive association of cardiorespiratory fitness with endothelial function [40,41] and a negative association of time spent exercising with arterial stiffness [42] as well as with wall thickness [43]. Furthermore, longitudinal studies have shown an attenuated decline of endothelial function with lifelong regular aerobic exercise until old age [5,44].

### 4.2. Clinical Vascular Effects of Short-Versus Long-Term Exercise

The extent and clinical relevance of vascular effects induced by single bouts of exercise and short-term interventions are idiosyncratic in a way. Although endothelial function appears to be modifiable within a few weeks of regular exercise [45], arterial stiffness and structural integrity of the vascular wall improve either slowly or not at all [46,47]. This is not in contrast to the consistent observations about beneficial vascular effects induced by long-term exposure to regular exercise but rather indicates the complexity of the underlying mechanisms. Therefore, the prediction of exercise-related improvements of future cardiovascular risk based on single biomarkers is limited, and therapeutic consequences should be drawn with caution.

#### 4.2.1. Influence of Age

Cardiovascular effects of chronic regular exercise are independent of sex and age [4,40] and are already apparent in minors and young adults [48], as demonstrated by the European Youth Heart Study [49] and the KiGGS study [50]. Considering the increasing prevalence of obesity and inactivity among children and adolescents [51], preventive efforts involving regular exercise should include the youngest individuals within a population. Additionally, facing the lack of an extended health span along with an increasing life expectancy, it is noteworthy that vascular adaptability towards exercise remains high until old age [52]. Thus, even the oldest individuals will considerably benefit from regular exercise, even if they have never exercised before.

#### 4.2.2. Influence of Cardiovascular Risk Factors and Diseases

Similar observations regarding clinical vascular effects of exercise have been made for both healthy individuals and patients with comorbidities, such as metabolic syndrome [40], arterial hypertension [53], and heart failure [40,54]. In contrast, exercise studies in animals provided evidence indicating an impaired mechanical transduction of elevated shear stress [55] and an attenuated increase in nitric oxide release [56] in the presence of cardiovascular risk factors such as arterial hypertension. As an example, the exercise-induced release of VEGF was attenuated in individuals with arterial hypertension, suggesting a blunted neocapillarization following exercise training [57]. Furthermore, microvascular dysfunction and reduced skeletal muscle oxygenation were associated with an exaggerated blood pressure response during exercise in newly diagnosed, untreated hypertensive individuals [58]. Nonetheless, vascular function in patients with cardiovascular risk factors or diseases remains highly adaptive to exercise, whereas structural vascular adaptations in the macrovascular system might be less likely to occur, compared to healthy individuals [59].

#### 4.2.3. Influence of Training Status and Physical Fitness Level

It is an appealing thought that cardiorespiratory fitness or previous training level influence the vascular adaptability to an exercise stimulus. It seems only logical that a highly trained organism does not have much room for further optimizations. Studies about endothelial adaptations to exercise in highly trained athletes support this in part, as some of them report better endothelial function than in untrained individuals [60,61,62], whereas many others report no differences or even lower function [63,64,65]. However, studies that compare vascular adaptations to exercise between trained versus untrained individuals are lacking. Furthermore, the interrelation of local structural and functional properties of the vascular wall is likely to be highly dynamic and influenced by training volume, intensity, or other factors [66]. Accordingly, endothelial adaptations to exercise in elite athletes may be highly dynamic and related to periodic variations of intensity, volume, frequency, and type of training during their season [40,65,67].

## 5. Exercise Training to Improve Vascular Fitness

### 5.1. General Aspects of Exercise Training to Improve Vascular Health

The World Health Organization recommends at least 75 min per week of vigorous physical activity or 150 min of moderate intensity [68]. The relationship between exercise and cardiovascular risk has been described in a curvilinear dose–response pattern [69], and numerous studies have consistently reported beneficial vascular effects of exercise training [5,42,43,44,59,70]. However, these effects depend on frequency, intensity, volume, and modality of the training, and are subject to considerable individual variation [71]. The F(requency)-I(ntensity)-T(ime)-T(ype) principle provides a practical approach to structure and adapt the training program according to the individual’s vascular responses (Figure 3).

### 5.2. F-I-T-T Principle: F(requency)

Longitudinal shear stress is a major stimulus for adaptive endothelial responses to exercise. Therefore, the total load of shear stimulus rather than the peak itself seems to be the critical determinant [72]. An animal study demonstrated that high frequencies of repeatedly elevated shear stress induce adaptive mechanisms within the endothelial cells [73]. Repeated elevations in shear stress could therefore explain some of the direct vascular effects of exercise.

Aerobic exercise training: A meta-analysis of randomized controlled exercise interventions in humans found no association between the frequency of sessions and improvement of endothelial function with aerobic exercise training [71]. Frequently repeated shorter bouts of exercise training might still be a considerable option, especially in individuals in whom the risk of injuries limits the application of high intensities or loads.

Resistance exercise training: A positive association between the frequency of resistance training sessions and improvement of endothelial function has been shown [71]. As with aerobic exercise training, frequently repeated shorter bouts of exercise training might be a considerable option in resistance training as well to minimize injury risk. However, the optimal frequency of training stimuli is still unclear in both aerobic and resistance exercise training.

### 5.3. F-I-T-T Principle: I(ntensity)

There are only a few studies, that compared intensity-dependent vascular functional and structural effects of the same exercise modality.

Aerobic exercise training: One study conducted a 12-week aerobic, ergometer-based training intervention at mild, moderate, and high exercise intensity [74]. Interestingly, only moderate but not high intensity led to improvements of nitric oxide-dependent endothelial function and less oxidative stress. The authors concluded that high intensities of aerobic exercise might induce massive acute oxidative and inflammatory stress, potentially attenuating favorable effects of elevated shear stress. Consistently, the short-term decrease in vascular function immediately after an exercise bout becomes larger with increasing exercise intensities [75]. In contrast, very short bouts of near-maximal- or supramaximal-intensity exercise have been demonstrated to induce similar [76] or even higher [77] vascular effects compared with traditional moderate-intensity aerobic exercise. The underlying mechanisms of this observation are not clear, but some evidence points towards an overexpression of mitochondrial enzymes and, thus, an increased time to fatigue during exercise [78], a stronger reduction in proinflammatory metabolites (i.e., oxidized low-density lipoprotein) in the long term [79], and a higher expression of antioxidant enzymes (i.e., glutathione peroxidase) [80]. Considering that the total load of shear stimulus rather than the peak itself seems to be the critical determinant of endothelial adaptation to exercise [72], a prolonged moderate-intensity workout might be preferred over a shorter high-intensity session to improve vascular function. Having in mind the greater injury risk associated with high-intensity training, more research is necessary to understand its potential beneficial effects in vascular structure and function.

Resistance exercise training: For resistance training, little evidence exists about the relationship between intensity and vascular adaptations, and it does not suggest a dose–response relationship between resistance exercise intensity and improvements in vascular function [71].

### 5.4. F-I-T-T Principle: T(ime)

Aerobic exercise training: The relationship of aerobic exercise with functional and structural vascular adaptations has been described in a dose–response pattern, with higher volumes of at least moderate-intensity exercise inducing higher functional vascular improvements [81]. However, the nature of this relationship is unclear, and no data exist about a minimum necessary time of training or a ceiling effect. To date, no conclusion can be drawn on the appropriate duration of a single session and, therefore, the appropriate volume of elevated shear stress that would provide the optimal stimulus for endothelial adaptation. One study found a higher increase in nitric oxide and a decrease in endothelial microparticles in the plasma after 40 min of moderate-intensity aerobic exercise compared to 20 min [82]. In contrast, one meta-analysis including aerobic exercise interventions longer than 4 weeks of duration found no influence of session duration (20–60 min) on changes of endothelial function independent of the overall training load [71]. Whether single training sessions should have a longer or shorter duration to attain the best possible stimulus for vascular adaptations cannot be answered now. However, as long as an adequate total training volume is guaranteed, the balance between session length and frequency may be chosen according to individual preferences of the trainee or general aspects of the training plan, such as subsequent time required for recovery.

Resistance training: Very limited data exist on the relationship of resistance exercise duration with functional and structural vascular adaptations. Therefore, no conclusions can be drawn to date.

### 5.5. F-I-T-T Principle: T(ype)

Continuous versus interval aerobic exercise training: High-intensity interval training (HIIT) is a frequently used method of improving fitness. A meta-analysis examining the effects of HIIT relative to moderate-intensity continuous training (MICT) on vascular function reported that HIIT was more effective at improving macrovascular (brachial artery) function than MICT. However, the authors reported that the variability in secondary outcome measures and the small sample sizes in the studies included limits this finding. Nevertheless, that review suggested that four intervals of 4 min (4 × 4 HIIT) at 85–95% of maximum or peak heart rate (HRmax/peak) interspersed with 3 min of active recovery at 60–70% HRmax/peak three times per week for 12–16 weeks is a type of exercise to enhance vascular function [83]. Another study [17] showed that a 4-week HIIT program was superior to MICT for improving macrovascular function (as assessed by brachial artery flow-mediated dilation) but not arterial stiffness (as assessed by pulse wave velocity). Over 12 months, changes in vascular function and arterial stiffness were similar for HIIT and MICT. In accordance, a meta-analysis including studies with exercise training interventions >4 weeks did not show any significant difference in the improvement of central arterial stiffness between HIIT and MICT [84]. It should be mentioned, however, that the results of studies comparing the two modes of aerobic exercise should be interpreted with caution, as in many studies, the HIIT and MICT training was not performed with similar effort and physiological strain. Controlling for the overall physiological stress exerted during the training sessions is important when comparing the two exercise modes [85,86].

Aerobic, resistance (dynamic/isometric), and combined exercise training: Both aerobic and resistance exercise seem to be associated with idiosyncratic patterns of blood flow and shear stress, which lead to distinct effects on arterial function and remodeling [20]. One study found similar antegrade but different retrograde shear patterns in response to aerobic and resistance exercise bouts [87], indicating different mechanisms of endothelial stimulation. This observation may at least in part explain why both local and systemic endothelial function show greater improvement after aerobic exercise compared to resistance training [71]. In fact, repeatedly marked blood pressure elevations during resistance training might reduce central arterial compliance [88,89]. As endothelial function was not reduced in these studies, the clinical implications of this structural remodeling associated with resistance training are unclear. However, it seems likely that they are independent of an age-related decline in structural and functional vascular capacities. Although aerobic exercise most likely is the stronger modifier of endothelial function, resistance training also seems beneficial. For example, one recent study showed protective alterations of cerebral blood flow after three months of resistance training [22].

Isometric exercise training has also been suggested as an effective intervention for inducing favorable vascular adaptations. In detail, a recent cross-over study showed that 4 weeks of isometric training (wall squats) reduced arterial stiffness (as suggested by a reduction in the augmentation index, derived from pulse-wave reflection) and total peripheral resistance [90]. In contrast, another study [91] found an improvement of microvascular function (as assessed by peak blood flow during reactive hyperemia) with dynamic resistance training (eight exercises, 50% of maximal repetition, three sets until moderate fatigue, for 10 weeks, three times/week) and not isometric handgrip training (at 30% of maximum voluntary contraction, four sets of 2 min). This later study did not find any additional improvements of combined exercise training compared with dynamic resistance training in vascular function. Differences between studies and exercise modes might be explained, at least in part, by the different amount of muscle mass involved during the exercise sessions and the type of blood vessels (micro- or macro-) examined.

All exercise modalities (aerobic, resistance, or combined) have been reported to induce improvements in vascular function depending on the characteristics of the exercise stimulus [71]. Because of distinct adaptations not only in the blood vessels but also in the skeletal muscles and other body systems in response to both aerobic and resistance exercise, training should involve both aerobic and resistance training modalities.

### 5.6. Individualization

Despite considerable scientific efforts, the mechanisms by which protective effects of exercise contribute to the maintenance and improvement of vascular health are still not fully understood. In consequence, evidence is lacking when it comes to the choice of optimal training programs for the individual [20,92]. Furthermore, patients’ adherence to exercise-based therapies is often undermined by the high efforts regular exercise requires [93,94]. Individuals with the poorest functional capacity who carry the highest risk of early frailty and disability, especially, do not seem reachable in many multifactorial risk-based studies [95]. By providing some variations to F-I-T-T in a patient-centered, flexible, and individual approach, training may be better tolerated, physically and mentally, than traditional linear training methods. Therefore, methods like nonlinear periodized exercise may promote patients’ adherence to exercise therapy while being equally effective as high-intensity or moderate-intensity continuous training [6,7,96,97].

Furthermore, sex differences in the training adaptations of the vessels have been reported [98,99]. Specifically, although in healthy, exercise-trained adults, large-elastic-artery-stiffening progression was attenuated, and exercise interventions were shown to improve arterial stiffness in sedentary middle-aged and older men and postmenopausal women; regular aerobic exercise was reported to improve endothelial function in men (by reducing oxidative stress and preserving NO bioavailability) but not to do so consistently in estrogen-deficient postmenopausal women. Thus, potential sex differences should also be taken into account when designing an exercise program.

## 6. Conclusions and Practical Implications

### 6.1. Conclusions

Exercise-induced hemodynamic changes lead to mechanical stress of the vascular wall, the release of circulating growth factors from the endothelium, and the release of exerkines from the exercising skeletal muscle and other organs. These three main adaptive stimuli lead to an increased activity of several molecular pathways within the vascular endothelial and smooth muscle cells, culminating in a better vasodilator and -constrictor responsiveness, reduced arterial stiffness, arterio- and angiogenesis, higher antioxidative capacities, and reduced oxidative stress. Recent research revealed a potential role of enhanced mitochondrial biogenesis and mitophagy, substrate metabolism, and insulin sensitivity in the vascular smooth muscle cells for exercise-induced vascular adaptations.

Acute bouts of exercise induce different vascular responses than regular training. Also, vascular adaptability towards exercise is high at any age in both healthy individuals and patients with cardiovascular diseases. This emphasizes the usefulness of exercise-based vascular health prevention from children to centenarians. Although athletes’ arteries seem to have higher functional capacities and less pathological structural remodeling than those of nonathletes, clinical biomarkers often fail to demonstrate significant differences, at least at a younger age. This puts the discussion on the table whether an already healthy vascular organ can be improved by exercise to a superhealthy level or not.

### 6.2. Practical Implications

Current evidence about specific aspects of exercise training, such as F-I-T-T, is limited, and exact training recommendations cannot be given. However, some practical implications can be extracted:

Some vascular adaptations, such as a favorable balance of angiogenesis and angiostasis as well as of vasodilator and vasoconstrictor responsiveness, require regular exercise, ideally throughout the entire life span. Therefore, individualization of exercise according to objective and subjective factors should be sought to achieve the best possible long-term training adherence.Repeated stimuli, at least every other day at initial stages and progressively increasing to 5–7 days per week, might be necessary to use the full potential of favorable physiological alterations, such as elevated blood pressure and improved glycemic control, which last for about 24 h post exercise.The cumulative volume of elevated shear stress seems more important than peak shear stress in terms of stimulating vascular remodeling. Thus, prolonged moderate-intensity workouts may be favored over shorter sessions with very high intensities, especially if injury prevention and long-term training adherence are important.High-intensity interval training may have additional benefits in the long-term, such as increased antioxidative and metabolic capacities, and thus should also be part of vascular exercise training.Resistance and aerobic exercise induce distinct macro- and microvascular adaptations; thus, both types of exercise should be implemented in comprehensive training for optimal vascular health.

## Figures and Tables

**Figure 1 cells-12-02544-f001:**
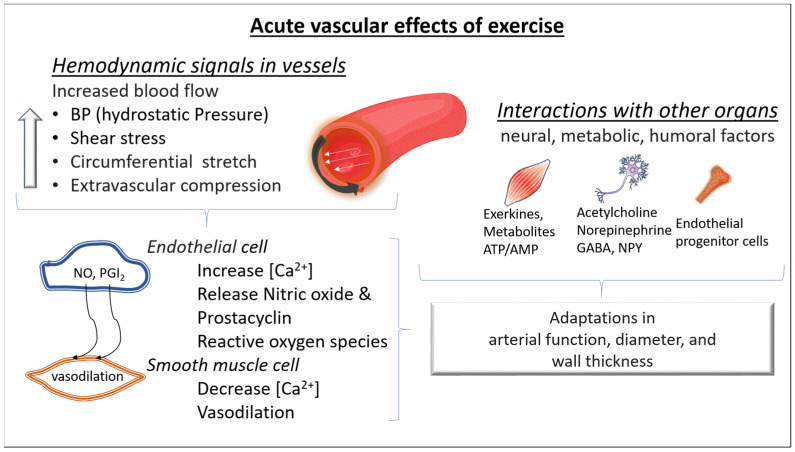
Acute molecular effects of exercise on blood vessels. During exercise, increased arterial blood flow leads to an elevation of blood pressure (greater hydrostatic pressure), luminal shear stress, and arterial wall stress. The consequence is a vasodilation predominantly in the resistance arteries (mainly due to a greater release of nitric oxide (NO) and prostacyclin (PGI2) from the endothelial cells and dilation of the smooth muscle cells). In addition, during exercise, changes in neural, metabolic, and humoral factors take place in both the micro- and macrovascular circulation. All these changes contribute to acute adaptations in arterial function, diameter, and wall thickness. NO: nitric oxide, PGI2: prostacyclin I2, NPY: neuropeptide Y, GABA: Gamma-aminobutyric acid, ATP/AMP: Adenosine triphosphate/Adenosine monophosphate; [Ca^2+^]: intracellular calcium.

**Figure 2 cells-12-02544-f002:**
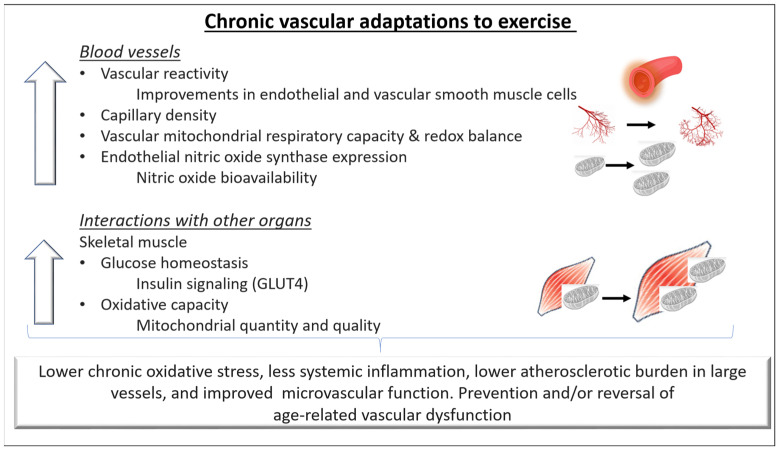
Chronic adaptations of blood vessels to exercise. Repeated hemodynamic stimuli due to regular exercise training can lead to better compliance and regulation of blood pressure in conduit arteries as well as in the small vessels. In addition, the vascular adaptations lead to a better capacity for oxygen delivery and diffusion from the capillaries to the skeletal muscle cells and possibly contribute to improvements in exercise capacity. Regular exercise can prevent and/or reverse age-related endothelial dysfunction in both the macro- and microvascular circulation by increasing Nitric oxide (NO) bioavailability, improving intracellular redox balance and mitochondrial health, and lowering systemic inflammation. Overall, these adaptations can reduce the atherosclerotic burden in large vessels and favorably alter the microvascular function in healthy individuals and in individuals with cardiovascular disease. GLUT4: Glucose transporter type 4.

**Figure 3 cells-12-02544-f003:**
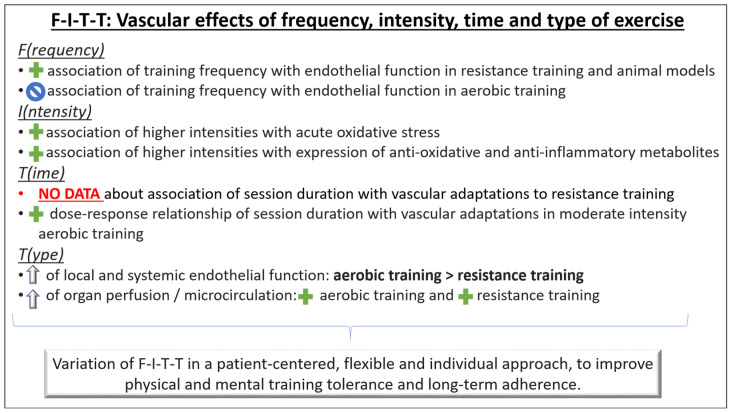
Principles of exercise training to improve vascular fitness. The F-I-T-T principle (frequency, intensity, time, type) provides a practical structure for individual tailoring of cardiovascular exercise therapy in preventive health care settings. 
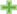
 association: Positive association; 
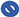
 association: No association; 
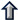
: Improvement.

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
