# Peer review of "Training the Vessels: Molecular and Clinical Effects of Exercise on Vascular Health—A Narrative Review"

_cells, 2023, doi:10.3390/cells12212544_

Round 1
Reviewer 1 Report
Comments and Suggestions for Authors
General comments:
This manuscript provides a comprehensive review of the molecular and clinical effects of exercise on vascular health. It is well written and organized appropriately. This reviewer has only a couple of very minor suggestions.
Specific comments:
1. Line 292: Insert “a” after “only”.
2. Line 366: Insert “been” after “have”.
Author Response
Reviewer #1:
This manuscript provides a comprehensive review of the molecular and clinical effects of exercise on vascular health. It is well written and organized appropriately. This reviewer has only a couple of very minor suggestions.
Authors’ reply:
The authors would like to thank the reviewers for the time and effort in improving this manuscript.
Reviewer’s comment:
Line 292: Insert “a” after “only”.
Authors’ reply:
Thank you, we have corrected the phrasing.
Reviewer’s comment:
Line 366: Insert “been” after “have”.
Authors’ reply:
Thank you, we have corrected the phrasing.

Reviewer 2 Report
Comments and Suggestions for Authors
Konigstein and colleagues present a review of the impact of exercise on vascular health with focus on both molecular mechanisms as well as clinical impact. Overall, the manuscript is comprehensive and clear. Comments are included below that may help strengthen the manuscript.
- The tail end of the abstract lists the major findings but lack cohesion. Please rewrite to better fuse this section (Ln 21-28)
- Line 36 – How is early vascular aging quantified to determine the 37% value? Please incorporate to better educate the reader to this topic area.
- As this review is focused on the impact of exercise on vascular health this reviewer would strongly suggesting shying away from cross-sectional studies, such as the information provided in Ln 124-135. Although these comparisons are interesting, a lack of definitive evidence that these adaptations arose from exercise vs hereditary/genetic factors cast doubts on this approach.
- Please clearly delineate vascular improvements specific to the macro- vs microvasculature as both regions serve some similar, but in most cases differing roles in blood flow regulation during exercise.
- Would suggest separating FITT principle headings for aerobic and resistance training to improve the reader’s interpretation of the current findings.
Author Response
Response to the reviewers (page numbers and lines refer to the track-change-version):
Reviewer #2:
Konigstein and colleagues present a review of the impact of exercise on vascular health with focus on both molecular mechanisms as well as clinical impact. Overall, the manuscript is comprehensive and clear. Comments are included below that may help strengthen the manuscript.
Authors’ reply:
The authors would like to thank the reviewers for the time and effort in improving this manuscript.
Reviewer’s comment:
- The tail end of the abstract lists the major findings but lack cohesion. Please rewrite to better fuse this section (Ln 21-28)
Authors’ reply:
Thank you, please find our corrections on page 1, lines 21-42: “Some vascular adaptations require regular exercise throughout the entire lifespan. Individuali-zation of exercise according to objective/subjective factors may promote long-term training ad-herence and injury prevention. Repeated stimuli 5-7 days per week might be necessary to use the full potential of favorable physiological alterations. The cumulative volume of elevated shear stress seems more important than peak shear stress. High-intensity interval training may have additional long-term benefits, thus, should be part of exercise training for optimal vascular health. Resistance and aerobic exercise induce distinct macro- and microvascular adaptations, thus, both types of exercise should be implemented in a comprehensive training regimen. Exercise increases shear stress on the vascular wall, and stimulates endothelial release of circulating growth factors, and of exerkines from the skeletal muscle and other organs. As a result, remodeling within the vascular walls leads to a better vasodilator and –constrictor responsiveness, reduced arterial stiffness, arterio- and angiogenesis, higher antioxidative capacities, and reduced oxidative stress. Although current evidence about specific aspects of exercise training, such as F-I-T-T, is limited, and exact training recommendations cannot be given, some practical implications can be extracted. As such, repeated stimuli 5-7 days per week might be necessary to use the full potential of these favorable physiological alterations, and the cumulative volume of elevated mechanical shear stress seems more important than peak shear stress. Because of distinct short- and long-term effects of resistance and aerobic exercise including higher and moderate intensities, both types of exercise should be implemented in a comprehensive training regimen. As vascular adaptability towards exercise remains high at any age in both, healthy individuals and patients with cardiovascular diseases, individualized exercise-based vascular health prevention should be implemented at any age from children until centenarians”.
Reviewer’s comment:
- Line 36 – How is early vascular aging quantified to determine the 37% value? Please incorporate to better educate the reader to this topic area.
Authors’ reply: Thank you, we added a definition of early vascular ageing (page 2, lines 49-52): “The prevalence of early vascular ageing in a population, i.e. defined as aortic pulse wave velocity >90th percentile, can be as high as 37% [1], which is why the maintenance of optimal vascular health throughout the entire lifespan is already accepted as a main goal of preventive health care [2].”
Reviewer’s comment:
- As this review is focused on the impact of exercise on vascular health this reviewer would strongly suggesting shying away from cross-sectional studies, such as the information provided in Ln 124-135. Although these comparisons are interesting, a lack of definitive evidence that these adaptations arose from exercise vs hereditary/genetic factors cast doubts on this approach.
Authors’ reply:
Thank you. More evidence provided by longitudinal studies on this issue is certainly necessary. However, there is some longitudinal evidence and also observational data comparing athletes with non-athletes and active versus non-active limbs in athletes, that in summary allow for the interpretations provided in this paragraph. We have adapted the phrasing in order to clarify (see page 4, line 138 – page 5, line 152): “Studies have also suggested that repeatedly elevated shear stress as it may occur during exercise training induces increases of the arterial luminal diameter [18]. In detail, larger conduit arteries (increased conduit artery size in epicardial arteries and those supplying skeletal muscle) have been reported in athletes compared to untrained healthy individuals [19]. Cross-sectional studies in athletes, as well as in individuals with high levels of phys-ical activity and aerobic fitness Between-limb studies in athletes have demonstrated higher arterial diameters and also suggest that regular training is associated with lower arterial wall thickness in the predominantly used limbs (carotid, femoral, popliteal- and brachial- intima–media thickness) [18, 20, 21]. Peak As a result, peak limb blood flow responses were found enhanced, also in comparison with non-athletes subjects in athletes suggesting greater vasodilator reserves in athletes and trained individuals that resistance arteries can also undergo increases in total cross-sectional area [19]., due to the repetitive episodic increases in arterial shear stress which elicit endothelium-mediated remodeling. The described structural vascular adaptations indicate greater vasodilator reserves in athletes and trained individuals.” Please be also aware that we substituted references [18], [20] and [21] by more suitable articles.
Reviewer’s comment:
- Please clearly delineate vascular improvements specific to the macro- vs microvasculature as both regions serve some similar, but in most cases differing roles in blood flow regulation during exercise.
Authors’ reply:
Thank you. Unfortunately, entirely separating micro- and macrovascular adaptation towards exercise won’t be possible at this stage of the process. However, we would like to raise your attention towards several paragraphs in the current manuscript, that already emphasize different effects of exercise in the macro- and the microvascular system:
Page 2, lines 65-67: “As a result, endothelial vasodilator and -constrictor responsiveness as well as pulse wave velocity and arterial stiffness are improved [9]. In addition, exercise induces vascular structural adaptations, and stimulates angiogenesis and arteriogenesis.”
Page 3, lines 94-95: “All these changes contribute to acute adaptations in arterial function, diameter, and wall thickness.”
Page 4, lines 125-127: “Overall, these adaptations can reduce atherosclerotic burden in large vessels and favorably alter microvascular function in healthy individuals and in individuals with cardiovascular disease.”
Page 4: lines 128-129: “Briefly, exercise stimulates angiogenesis, the formation of capillary networks, and arteriogenesis, the growth of preexistent collateral arterioles.”
Page 7, lines 261-266: “As an example, the exercise-induced release of VEGF was attenuated in subjects with arterial hypertension, suggesting a blunted neo-capillarization after exercise [57]. Furthermore, microvascular dysfunction and reduced skeletal muscle oxygenation were associated with an exaggerated blood pressure response during exercise in newly diagnosed, untreated hypertensive individuals [58].”
We further rephrased several sentences in order to improve clarity about where in the vascular system specific effects of exercise occur and added some more information on the type of vessels examined:
Page 3, lines 87-96: “During exercise increased arterial blood flow increases leadingleads to an increased elevation of blood pressure (greater hydrostatic pressure), an increase in luminal shear stress and increased arterial wall stress. The consequence is a,and consequently vasodilation predominantly in the resistance arteries (mainly due to a greater release of nitric oxide and prostacyclin from the endothelial cells and dilation of the smooth muscle cells). In addition, during exercise, changes in neural, metabolic, and humoral factors take place in both, the micro- and macrovascular circulation. All these changes contribute to acute adaptations in arterial function, diameter, and wall thickness. NO: nitric oxide, PGI2: prostacyclin, NPY: neuropeptide Y.”
Page 4, lines 122-125: “Regular exercise can prevent and/or reverse age-related endothelial dysfunction in both, the macro- and microvascular circulation, by increasing NO bioavailability, improving intracellular redox balance and mitochondrial health, and lowering systemic inflammation.”
Page 5, lines 174-178: “In addition to the direct vascular effects of exercise these macrovascular effects of exercise, exerkines (signaling molecules such as proteins, metabolites, and extracellular vesicles) are released in response to acute and/or chronic exercise from the working skeletal muscle, the cardiometabolic, nervous and immune system and induce adaptations in the microvascular system [26].”
Page 7, lines 266-268: “Nonetheless, vascular function in patients with cardiovascular risk factors or diseases remains highly adaptive to exercise, whereas structural vascular adaptations in the macrovascular system might occur less likely, compared to healthy individuals [59].”
Page 10, lines 377-380: “A meta-analysis examining the effects of HIIT relative to moderate-intensity continuous training (MICT) on vascular function, reported that HIIT was more effective at improving macrovascular (brachial artery) function than MICT.”
Page 10, lines 385-387: “Another study [84] showed that a 4-week HIIT program was superior to MICT for improving macrovascular function (as assessed by brachial artery flow-mediated dilation), but not arterial stiffness (as assessed by pulse wave velocity)……….
Page 10, lines 387-390: “a meta-analysis including studies with exercise training interventions >4 weeks, did not show any significant difference in the improvement of central arterial stiffness between HIIT and MICT [85]”
Reviewer’s comment:
- Would suggest separating FITT principle headings for aerobic and resistance training to improve the reader’s interpretation of the current findings.
Authors’ reply:
Thank you for your comment. We understand the reviewer’s concern. Because the FITT principle is a holistic approach to exercise used for aerobic and resistance training, there is considerable overlap between those two modalities at some points, which somewhat complicates a strict separation. In order to follow your suggestion to improve reader’s interpretation, please find our adaptations between page 8, line 300 and page 11, line 431. We have added a separate paragraph presenting the effects of dynamic versus isometric exercise (page 11, lines 413-419) and a paragraph comparing the effects of aerobic exercise, resistance training (dynamic vs isometric), and combined exercise training (page 10-11, lines 397-431). In addition, in the revised manscript we now present the effects of continuous versus interval aerobic exercise in a separate paragraph. We hope this is a solution you might appreciate.

Reviewer 3 Report
Comments and Suggestions for Authors
The author conceptually summarized that acute and chronic molecular vascular effects of exercsise and clinical effects of exercise in this review. Despite limited research on FITT, the author showed some useful clinical exercise guidline. However, i think it would have been better for the author to review the effects of exercise (molecular vascular effects of exercise et al.,) focusing on FITT for readers. The manuscript is very well organized.
* line 32: 1.1. Background. Please replace “1.1” with “1”.
* line 210~211: please check the sentence.
Author Response
Reviewer #3:
The author conceptually summarized that acute and chronic molecular vascular effects of exercsise and clinical effects of exercise in this review. Despite limited research on FITT, the author showed some useful clinical exercise guidline. However, i think it would have been better for the author to review the effects of exercise (molecular vascular effects of exercise et al.,) focusing on FITT for readers. The manuscript is very well organized.
Authors’ reply:
The authors would like to thank the reviewers for the time and effort in improving this manuscript.
In ther revised bersion of the mansucript we have added more information on the characteristics of the exercise program for improving vascular function. We are now presenting more information on the type of aerobic exercise (continuous/interval) and on the mode of exercise (Aerobic exercise, resistance (dynamic/isometric), and combined exercise training used in the different studies (Please see pages 8-11, lines 309-431) and inserted a new paragraph discussing the effects of “Continuous versus interval aerobic exercise training” (page 10, lines 376-395)..
Reviewer’s comment:
* line 32: 1.1. Background. Please replace “1.1” with “1”.
Authors’ reply:
Thank you, we have corrected the phrasing.
Reviewer’s comment:
* line 210~211: please check the sentence.
Authors’ reply:
Thank you, we have corrected the phrasing.
